# Biophysical Control of the Glioblastoma Immunosuppressive Microenvironment: Opportunities for Immunotherapy

**DOI:** 10.3390/bioengineering11010093

**Published:** 2024-01-18

**Authors:** Landon Teer, Kavitha Yaddanapudi, Joseph Chen

**Affiliations:** 1Department of Bioengineering, University of Louisville, Louisville, KY 40292, USA; landon.teer@louisville.edu; 2Department of Microbiology and Immunology, University of Louisville, Louisville, KY 40202, USA; 3Immuno-Oncology Program, Brown Cancer Center, Department of Medicine, University of Louisville, Louisville, KY 40202, USA; 4Division of Immunotherapy, Department of Surgery, University of Louisville, Louisville, KY 40202, USA

**Keywords:** glioblastoma, immunotherapy, tumor microenvironment, biophysics

## Abstract

GBM is the most aggressive and common form of primary brain cancer with a dismal prognosis. Current GBM treatments have not improved patient survival, due to the propensity for tumor cell adaptation and immune evasion, leading to a persistent progression of the disease. In recent years, the tumor microenvironment (TME) has been identified as a critical regulator of these pro-tumorigenic changes, providing a complex array of biomolecular and biophysical signals that facilitate evasion strategies by modulating tumor cells, stromal cells, and immune populations. Efforts to unravel these complex TME interactions are necessary to improve GBM therapy. Immunotherapy is a promising treatment strategy that utilizes a patient’s own immune system for tumor eradication and has exhibited exciting results in many cancer types; however, the highly immunosuppressive interactions between the immune cell populations and the GBM TME continue to present challenges. In order to elucidate these interactions, novel bioengineering models are being employed to decipher the mechanisms of immunologically “cold” GBMs. Additionally, these data are being leveraged to develop cell engineering strategies to bolster immunotherapy efficacy. This review presents an in-depth analysis of the biophysical interactions of the GBM TME and immune cell populations as well as the systems used to elucidate the underlying immunosuppressive mechanisms for improving current therapies.

## 1. Introduction

Glioblastoma (GBM) is the most aggressive and common type of primary brain cancer. The median patient survival is only 15 months, and the 5-year survival of 5–7% remains one of the lowest among cancer types [1,2,3,4,5]. This poor prognosis remains despite aggressive treatment via surgical resection, radiotherapy, and chemotherapy [6,7]. GBM progresses as a result of pro-malignant physical and chemical factors in the TME that promote therapy resistance, immune evasion, and rapid tumor dissemination throughout the brain. This ultimately leads to an incomplete elimination of the GBM cells, which seed and generate more aggressive, secondary tumors [8,9,10,11] that reduce survival and quality of life [11,12,13]. Unfortunately, the current therapeutic approach has not been effective, and research into alternative options is desperately needed [14,15].

Immunotherapies represent a paradigm shift in cancer therapy, leveraging an individual’s own immune cells to eliminate the cancer cells. This has proven to be effective in a wide range of cancer types and provides hope for GBM treatment [16,17,18,19,20,21]. In a healthy human body, the host’s immune system will naturally work to eliminate cancer cells, most often via natural killer (NK) cells [22,23,24,25,26] and T cells [24,26,27,28,29], but macrophages have also been implicated in possessing a small tumoricidal capacity, although they more often adopt a protumorigenic phenotype [26,30]. These cells utilize many pathways and mechanisms to eliminate cancerous cells before they aggregate and develop a tumor. Despite the overall efficiency of these mechanisms, some cancer cells can shift their phenotypes, release immune-cell-suppressing signals, and alter the surrounding extracellular matrix (ECM) to increase tumor survival. Thus, the basic rationale for immunotherapy is to equip immune cells with new molecular tools to identify and successfully eradicate cancerous cells. However, despite the overwhelming success of immunotherapy, its effectiveness is not universal and depends largely on the TME of different tumors.

Tumors such as GBM are labeled as immunoevasive or “cold tumors” and have less immune cell infiltration [31,32,33]. Such changes associated with GBM include the adaptation of surface receptors to limit immune cell binding [34,35], the reprogramming of immune cells such as parenchymal macrophages (microglia) and neutrophils to more protumorigenic profiles [36,37,38,39,40,41], and the recruitment of inhibitory immune cells such as regulatory T cells (Tregs) [42,43] and myeloid-derived suppressor cells (MDSCs) [44,45,46], which secrete immunosuppressive cytokines such as TGFβ, IL-10, and IL-35 [47,48,49]. Together, these factors reduce the amount of tumoricidal immune cell infiltration and generate protumorigenic/anti-immune signals to create a highly immunosuppressive microenvironment. Therefore, efforts to dissect these immunosuppressive mechanisms in the GBM TME are needed to enhance the infiltration potential of native immune cells [50,51].

The GBM TME is a complex milieu that contains biomolecular and biophysical signals that regulate several aspects of tumor progression. Recent research has revealed important immunomodulatory cytokines, secreted factors, and cell–cell crosstalk; however, emerging work has also identified biophysical signals such as ECM composition and stiffness as critical regulators of host immune responses [52,53,54,55]. Tumor biophysics play a crucial role in regulating cancer progression. Decades of studies have previously focused on increasing the understanding of the genetics and biochemistry that comprise the cancer-causing DNA known as “onco-genes” [56]. It is now appreciated that the TME continuously communicates with the tumor cells via cell–cell interactions, ECM binding, and the production of soluble factors [57,58]. These tumor-TME interactions can promote disease progression via increased DNA damage [59,60], leading to mutations that trigger the development of the hallmarks of cancer. Additionally, there is evidence that the biophysical TME may cause tumor cells to become more immunomodulatory, secreting immunosuppressive cytokines to turn off the immune system, and may also regulate immune cell activity as well [61,62]. This review plans to summarize the current understanding of the interactions between the biophysical TME and immune cell populations and describe the bioengineering models and tools used to elucidate the underlying mechanisms of these interactions to leverage them for immunotherapy applications to improve cancer treatment.

## 2. Biophysical Aspects of the TME

As previously mentioned, tumor cells are in constant communication with the TME. The biophysical signals in the TME are diverse and are transmitted to the cells and converted into intracellular signals through a process known as mechanotransduction. This section reviews some of the most common types of mechanical signals produced by the TME and discusses the impact on cells in the tumor.

### 2.1. ECM Composition

Solid tumors comprise a wide range of highly expressed ECM proteins that include laminins, fibronectin, elastin, and fibrillar collagen [63]. These proteins provide functionality and stability to the tumor environment for disease progression and accounts for up to 60% of the tumor mass [64]. Indeed, the presence of these proteins plays a key role in regulating the pro-invasion and therapeutic resistance within the tumor that help to direct cell migration, adhesion, and proliferation similar to that seen in early development [65]. Several ECM proteins of interest in GBM have been identified over the last several years including osteopontin, hyaluronic acid (HA), and laminin and are known to increase invasion potential through mechanotransduction signals transmitted throughout the cytoskeleton [66,67,68]. Further, HA is the primary ECM component in the brain, and it has been strongly implicated in GBM tumor development and depends on the HA molecular weight—high-molecular-weight HA has an anti-tumor effect and low-molecular-weight HA has a pro-tumor effect [67,69,70,71,72]. A higher initial presence and turnover of HA can contribute to greater CD44 binding by GBM cells, which can cause a shift to a more malignant phenotype via downstream processes. For these reasons, current and future culture systems should include biologically relevant ECM components to recapitulate physiologically relevant behaviors lost via standard in vitro platforms.

### 2.2. ECM Mechanics

Besides ECM composition, the mechanics of the ECM can direct cell behavior [73] and tumor malignancy. Increased substrate stiffness has been shown to increase proliferation and single-cell migration in GBM. ECM stiffness can also increase cell adhesiveness, which interestingly reduces collective migration due to dense networks of cell–substrate adhesions [74,75]. These stiffnesses also affect the density of the matrix and can facilitate primary tumor escape during metastasis [76] by providing permissive porous environments to increase cell escape from the tumor or inhibitory dense environments that would confine the cell to the tumor. Additionally, substrate stiffness can upregulate transcription factors such as Twist1, Snail, and SOX2, which increase the mesenchymal phenotype transition and cell stemness, respectively [77,78]. These qualities correlate with cancers that are more likely to metastasize, which reduces patient survival. Indeed, these bulk mechanical properties present biophysical cues that facilitate tumor progression.

### 2.3. Interstitial Fluid Flow

The increased mechanics of the intratumoral space caused by the high stiffness and density can also affect the flow of interstitial fluid throughout the tumor. The flow of interstitial fluid through the tumor exposes the cells to fluid pressure and soluble cues such as pro-angiogenic factors and anti-inflammatory TGF-β1 signals [79]. These factors function to inhibit anti-tumor immune cells and promote pro-tumor immune cells by maintaining the inflammation of the tumor to a sustainable level. Moreover, these signals can help to direct cancer cell migration [80] to a more favorable route for metastasis. The interstitial fluid flow in the tumor core contributing to the pressure differences in GBM is also characterized by a low pH and low oxygen concentration, which perpetuate necrosis and DNA damage [81]. This further drives exposed tumor cells towards more a malignant mesenchymal phenotype associated with cancer progression, and it increases the expression of the urokinase-type plasminogen activator and its receptor to promote cancer progression via increasing the migration capacity [82]. Truly, the repercussive anti-inflammatory signaling and DNA damage effects downstream of the increased interstitial fluid flow provide great synergy in progressing tumor development.

## 3. Biophysical TME-Immune Cell Interactions

It has been established that the biophysics of the TME and ECM can regulate cell behavior in a pro-tumorigenic capacity. However, besides cancer cells, other types of cells including the immune cell populations are regulated by the biophysical TME (Figure 1). This section summarizes the interactions between the biophysical cues of the TME and the immune cells and describes how the TME can limit the immune-cell-mediated tumoricidal activity.

### 3.1. Substrate Stiffness

Healthy and cancerous somatic cells can be characterized by their degree of mechanosensing and mechanotransduction. Generally, stiffer substrates correspond to greater cell migration due to increased contractility [83,84,85,86]; yet, for the immune cells, the effect is less pronounced with respect to cell migration (Figure 1A). Dendritic cells are a type of antigen-presenting cell that helps link the innate and adaptive immune systems by presenting an antigen to a lymphocyte such as a T cell to induce recognition and a response against the tumor cells [87]. During improper antigen presentation, the T cells are unable to effectively recognize and eliminate the source of the antigen. On stiffer substrates, it has been observed that dendritic cell functionality is negatively impacted, with Ref. [88] implying that dendritic cells within a stiffer TME are less capable of inducing a proper T cell response. Moreover, T cell functionality is also impacted directly by ECM stiffness as T cell migration is negatively correlated with tumor stiffness [89,90] and the anti-PD-1/PD-L1 resistance of cancers is promoted [89,91], thus reducing the tumoricidal capacity of the infiltrating CD8+ cells. This reduced tumoricide is further seen in more complex TME interactions involving the tumor cells. GBM propagates in a soft matrix that does not normally promote upregulated FAK activity [92], which can result in reduced MRFT-A localization [93]. Abundant MRFT localization has been demonstrated to sensitize other aggressive cancers to the T cell response; therefore, a soft matrix such as the brain would not only reduce T cell activation but also remove a targetable receptor [94]. Other immune cells such as the microglia, considered the macrophages of the brain, are known to preferentially migrate towards stiffer environments [95]; however, this proves to be detrimental as macrophages are one of the most reprogrammed types of cells and easily transition towards a protumorigenic phenotype known as tumor-associated macrophages (TAMs) [96,97,98]. Ultimately, the collective effect of increased substrate stiffness on different immune cells creates a unique spatial organization of immune cells that serves to best benefit the development of the tumor.

### 3.2. Intratumoral Pressure

In a similar manner to the stiff tumor, an interstitial fluid flow induces pro-tumorigenic effects on the immune cells in the TME (Figure 1B). Tumors often utilize the lymphatic system to invade and metastasize [99]. This system specializes in fluid drainage and, therefore, it is not uncommon to find heightened pressure gradients at the junction of tumor and lymphatics. These junctions experience an increased flow and mechanical stress, which serves to cause DNA damage and increase the production of cytokines such as TGF-β1 to reduce inflammation [99,100,101]. Furthermore, the interstitial pressure is sufficient to induce a pro-tumorigenic macrophage phenotype as seen by the upregulation of M2 macrophage markers ArgI, TGM2, and CD206 [102]. This development of M2 macrophages further works to inhibit T cell function via the additional TGF-β1 production and expression of immune checkpoint ligands that inactivate infiltrating T cells [103,104]. Moreover, the development of M2 macrophages can recruit regulatory T cells (Tregs) [105] that can further inhibit CD8^+^ T cell activity by binding CTLA-4 [106]. T cell proliferation and activity can be further reduced due to the upregulation of platelet-derived growth factor isotypes [107,108] expressed from GBM [109] which help to further regulate the intratumoral fluid flow [110]. Collectively, these findings demonstrate that an increase in intratumoral pressure works to promote the cancer development through increased damage which, in turn, leads to the secretion of anti-inflammatory cytokines that reduce the functionality of tumoricidal T cells via dysregulation of the T cell receptors (TCR). This dysregulation prevents proper mechanotransduction across the TCR from occurring, which limits T cell functionality and promotes tumor development. 

### 3.3. Confined Migration

Considering the density of the brain parenchyma, it is unsurprising that GBM is an innately aggressive cancer. For many cell types, migration in a confined environment can lead to DNA damage via nuclear rupture [104], which serves as the basis for the development of metastatic cancer [111]. During infiltration of the TME, the immune cells will enter a state of confined migration (Figure 1C). Macrophages have been observed to form a protective actin cortex that shields against compressive forces that can damage the nucleus and lead to cell death [112]. This enables a more efficient and safe form of migration necessary for patrolling throughout the confined environments. Likewise, T cells navigate a variety of confined spaces as they patrol throughout the body; however, while patrolling, the T cells also interact with antigen-presenting cells. This presents a balance that must be maintained between migrating quickly through an environment and spending enough time in the same location as the antigen-presenting cell to become activated against an antigen [113]. Moreover, dendritic cells face a similar situation in that while immature dendritic cells quickly navigate confined spaces in search of antigens [114,115], there is an inverse relationship between antigen uptake efficiency and cell speed [116]. The immune cells will normally receive a reprieve from confinement as they patrol the body, but the brain parenchyma provides no such relief. Interestingly, while neutrophils are not abundant in the GBM TME [117,118], their recruitment to the tumor site has been well studied. IL-8 and CXCL-1 attract neutrophils to the dense TME where the cells become clustered and immobilized. The confined cells secrete additional IL-8 which promotes tumor cell escape by disrupting cell junctions to increase the permeability of endothelial cells in the brain [119,120,121]. From these findings, it is established that while the immune cells can navigate confinement without experiencing excessive cell damage, the increased mutation rate of cancer cells coupled with the reduced duration of interaction with the immune cells contribute, in part, to an inefficient system of immune cell activation which limits tumoricidal functionality.

## 4. Bioengineering Systems

To study the adverse interactions between the TME and immune cells that lead to immunogenic reprogramming to a pro-tumorigenic state, researchers have developed bioengineering platforms to elucidate the underlying mechanisms utilized by the TME. Understanding these mechanisms enables the recapitulation of the in vivo environment and allows for the discovery of novel, targetable axes for immunotherapies that can be used to overcome the current limitations of immunotherapies. This section describes the tools currently used and the implications they have on developing new immunotherapies for GBM.

### 4.1. Biomaterial Systems

Biomaterials are synthetic or natural materials that can be engineered to mimic physiological and pathological environments and are foundational to investigate cell-ECM interactions. Among the most common biomaterials are hydrogel-culture-based systems due to their versatility in controlling factors such as stiffness, ECM composition, and other biomolecules such as growth factors [122]. In the context of solid tumors such as GBM, the most important immune cell functions to consider are tumor infiltration and tumor cell elimination. Hydrogel-based platforms have proven vital in dissecting the interplay between immune cells and the tumor ECM. The mechanosensing capacity of T cells has been interrogated via TCR-mediated activation through the introduction of HA binding [123] and the tuning of the stiffness to biomimetic levels [123,124]. From these studies, the importance of the TCR in transmitting the mechanical stimulus from the substrate or cancer cell to the immune cell body is elucidated and identified as a promising axis for generating pro-T cell-activity systems that can better eliminate cancer cells. Adjusting the oxygen levels through cross-linking fibrinogen to fibrin has been used to recreate physiological conditions contributing to immunosuppressive cues experienced by immune cells as they invade the hypoxic tumor regions [125,126]. In addition, macrophage polarization can be induced via peptide conjugation to pro-tumorigenic and anti-tumorigenic states [127,128], demonstrating a degree of tunability that can be utilized in directing immune cells to a more favorable, anti-tumor phenotype. Ultimately, these systems work to recapitulate the in vivo tumor system to promote a more biologically accurate response in the cancer and immune cells by introducing culture platforms that provide cell–cell and cell-ECM physical interactions lacking in traditional culture platforms.

### 4.2. Microfabrication

Microfabrication approaches such as 3D printing and microfluidics represent important tools for studying the contribution of geometric cues as well as observing cell phenomena in a single-cell context. Many studies leverage these tools to decipher the migration mechanisms of immune cells. In the 3D context, CD8+ T cell amoeboid migration operates through a contractility driven mechanism via RhoA activation that is unique to the 2D environment [129,130,131]. Additionally, the T cell transfection efficiency can be modified on microfluidic systems through mechanoporation via stretching to engineer T cells with greater motility, antigen recognition, and antigen elimination [132,133]. Furthermore, microfluidic platforms are useful for observing immune cell phenomena that may be lost or otherwise impossible to view in traditional in vitro experiments. T cells have demonstrated phenotypic shifts correlating to unique locomotion mechanisms dependent on ECM fiber alignment [134]. Further, microfluidic platforms have been used to isolate and observe single T cell and antigen present cell (APC) interactions for elucidating Ca^2+^ mobilization dynamics in early T cell activation cascades that regulate cytokine production [135]. These interactions rely on the ability of the APC to process the antigen and present it to the T cell to induce a cytolytic response against the cancer cells, a process that was improved upon by using microfluidic squeezing to induce pore formation in several non-APC cells that could then activate T cells with greater efficiency than in traditional activation processes [136]. While these cell behaviors can be isolated to and observed in simple devices, the greatest strength comes from confining multiple cells to small regions on a single platform, allowing for the observation of many independent cell functions.

### 4.3. Advanced In Vitro Systems

Creating more complex, biologically relevant in vitro systems serves as an intermediate step before reaching in vivo models. Co-culture models combine multiple cell types into a single culture environment to analyze real-time interactions between the cells, and lab-on-a-chip technology is useful for combining multiple physiological systems to determine how cell processes differ between the environmental changes. Macrophages and microglia are normally susceptible to reprogramming via GBM cues that promote tumor development; yet, when cultured separately, these changes cannot be observed.

Therefore, co-culture systems allow for the necessary paracrine and juxtacrine interactions (ephs/ephrins [137] or P-selectin/PSGL-1 [138]) involved in the bidirectional communication between the tumor and immune cells to occur and be observed. These interactions are intended to help recognize the cancer cells for immune-cell-mediated elimination but are often dysregulated to instead promote tumor survival and development. For instance, M2 macrophages have been suggested as positive prognostic markers in lung cancer due to the development of CD204+/CD68+ receptor profiles that can reduce tumor progression [139]—these interactions can only be studied through co-culture models. Furthermore, when cell culture becomes inaccessible, lab-on-a-chip technology provides a favorable alternative. GBM-on-a-chip platforms provide systems containing the biological and chemical units of a tumor that are involved in suppressing immune cell function [140]. Tumor heterogeneity represents a difficult barrier to bypass in developing effective immunotherapies; however, lab-on-a-chip technology provides a system that can be tailored to easily and accurately generate the heterogeneous regions that can be found in tumors in vivo in terms of their ECM components and characteristic 3D geometry. Among these regional characteristics are substrate stiffness, CO_2_ concentration, fluid flow, etc. [140]. After using the traditional culture platforms and tunable platforms to assess specific mechanisms, lab-on-a-chip technology can combine multiple parameters to test whether a potential synergy exists and aid in the development of drug screening tools or a deeper study of cancer immunoevasion.

## 5. Immunotherapy Applications

From the mechanistic insights gained through bioengineering systems, researchers can leverage these interactions to optimize immunotherapy protocols to increase efficacy and improve patient survival. By engineering the culture expansion platforms of immune cells and the activities of immune cells, efforts to increase the tumoricidal capacity of immune cells are growing. In recent years, many types of cancers have seen significant improvement due to the development of novel immunotherapies, and this section lists some of the most significant therapies, their developments in treating GBM, and the biophysical design of the therapies (Figure 2).

### 5.1. Tunable Hydrogel Culture Systems

ECM composition and mechanics are involved in determining the degree of functionality of not only cancer cells but immune cells as well. For GBM treatments, studies have been conducted that utilize HA due to the ease with which it can be modified without sacrificing material characteristics [142,143]. This degree of tunability has proven useful by helping to expand and activate T cells that are otherwise limited by the relatively slow growth and loss of characteristic phenotypes in traditional culturing methods (Figure 2A). HA gels are easily conjugated with signals necessary to activate T cells in greater magnitudes than with soluble signal presentation [141]. In addition, T cells demonstrated varying degrees of TCR activation depending on substrate stiffness, suggesting that a mechanical signal distributed across the TCR may be needed [141]. The benefits of HA culture systems are not limited to T cell activation, as microglia are also impacted through binding interactions. Specifically, the presence of high-molecular-weight HA is inversely related to the functionality of microglia [144,145]. Microglia function as the brain’s resident macrophage population, so it is vital in ensuring a homeostatic environment through neuronal circuit pruning and ECM remodeling [146,147,148]. In the presence of healthy brain tissue, there is minimal HA degradation that warrants microglial activity, but in GBM, the rapid expansion of cells causes tissue damage which would attract microglia to the tumor site where they can transform into TAMs. However, using ex vivo cultures, the microglia can be trained and transitioned to a more tumor-inhibitory phenotype through exposure to high-weight HA. Thus, these tunable HA systems, which incorporate mechanical signals and ECM signals, can be leveraged to optimize immune cell expansion and activity and can be integrated into immunotherapy workflows.

### 5.2. Chimeric Antigen Receptor Therapy

In addition to the removal and reintroduction of cells following the ex vivo differentiation into an anti-tumor phenotype, the direct modification of immune cells has also had success in treating various types of cancers. The generation of chimeric antigen receptors (CARs) for T cells involves removing T cells from a patient and using gene engineering to induce the production of a single type of synthetic receptor for antigen recognition [149,150]. To increase successful T cell transfection, microfluidic systems have stretched cells to increase membrane pore formation, which allows for greater mRNA entry to generate the CAR (Figure 2B) [132,133]. This bypasses the need for TCR binding by directly focusing on antigen receptor binding [151], which could limit the mechanosensing capacity of the CAR T cells if the antigen receptor is less mechanosensitive. For non-solid tumor cancers (leukemia, lymphoma, and myeloma), this has not proved consequential, as tumor penetration is not an issue. When considering CAR T for solid tumors, the therapy has been less effective due to the lack of tumor infiltration and immunosuppressive TMEs [152,153,154]. This stems from several previously mentioned factors including ECM density and immunosuppressive receptor expression by tumors such as GBM. Further, other limitations include the overwhelming quantity of inflammatory cytokines released by expanded CAR T cells during cytokine release syndrome [154,155]. To combat this effect, and potentially make the treatment more viable in immune-sensitive locations such as the brain, OFF switches have been added to the T cells to provide a quick and reversible means to prevent the detrimental effects [156], but these remain preliminary and unused in solid tumors at this time. Interestingly, the development of CAR T technology has recently been applied to macrophages and could be used to bypass the immune reprogramming sequence that converts them into TAMs and instead promote phagocytosis. Moreover, the amount of engulfment correlated with increased target stiffness [140,156,157,158,159], suggesting a prominent role of biophysics in immune cell processes that have not yet been well studied. The use of bioengineered tools to increase CAR T transfection efficiency and the importance of receptor biophysics on T cell tumoricidal activity support the inclusion of biophysical parameters in the development of next-generation CAR T technologies.

### 5.3. Checkpoint Blockade

Another therapy option utilized in even more cancers is the use of immunologic checkpoint blockades. This therapy uses antibodies to inhibit the programmed death of T cells induced by the binding of PD-1 or to prevent the activation of the T cell via CTLA-4 [160,161,162]. These receptors naturally become activated to prevent T cell overactivity; however, cancer cells can trigger these pathways as well to limit T cell-mediated tumor killing through the reduction in cytokine and granule production. To prevent this inhibition, antibodies are developed that are specific to either the tumor cell (PD-L1) or the T cell (PD-1/CTLA-4) and are administered [160,161,162,163]. The type of antibody used and its receptor target are variable depending on the state of the tumor and the effectiveness in promoting T cell activity. Moreover, the binding of these receptors induces a tensile force in the T cell similar to that seen in ECM-integrin binding [164]. While the implications of these binding interactions have not yet been explored, TCR binding has been implicated to be mechanosensitive. Given this T cell mechanosensitivity, it would not be surprising to find that checkpoint blockade therapies may benefit from mechanically modifying the antibody conjugate to induce greater T cell activation and proliferation via the inhibition of PD-1 and CTLA-4.

## 6. Conclusions

GBM is an extremely deadly type of brain cancer that has remained difficult to treat in part due to its high resistance to standard therapies and immunotherapy. The dynamic TME is a key driver of cancer progression and presents pro-malignant biophysical signals including ECM composition, density, stiffness, and interstitial fluid flow, which helps direct the flow of soluble cues, increases the intertumoral pressure on the cells, and generates hypoxic regions. Ultimately, these factors work together to promote tumor growth and disease progression; however, these factors also dysregulate immune cells through the production of immunosuppressive cytokines, the recruitment of inhibitory immune cells, and/or the conversion into tumor-supportive tumor-associated immune cells. Biomedical research has made strides in elucidating the underlying mechanisms of the biophysical TME on immunosuppression through novel bioengineered tools and platforms. The insights gained have been eagerly applied to bolster immune cell expansion techniques and enhance immunotherapy through the biophysical manipulation of the ECM and immune cell receptor manipulation. From this work, future research seems well poised to generate novel strategies that can overcome the barrier of immunosuppression and increase patient survival in GBM patients.

## Figures and Tables

**Figure 1 bioengineering-11-00093-f001:**
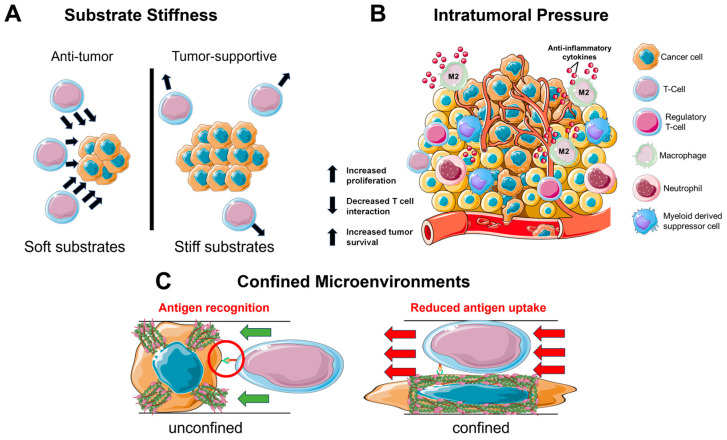
Biophysical properties reduce immune cell–cancer cell interactions. Within the TME, many biophysical cues work together to create an immunosuppressive environment. Substrate stiffness reduces the migration capacity of T cells and limits the tumoricidal capacity of CD8+ cells (**A**); intratumoral pressure transforms macrophages into a tumor-supportive phenotype that produces anti-inflammatory cytokines that inactivate infiltrating T cells and recruits regulatory T cells (**B**); and dense tumor tissue alters cell shape and cytoskeletal organization and limits the chance of antigen recognition (**C**).

**Figure 2 bioengineering-11-00093-f002:**
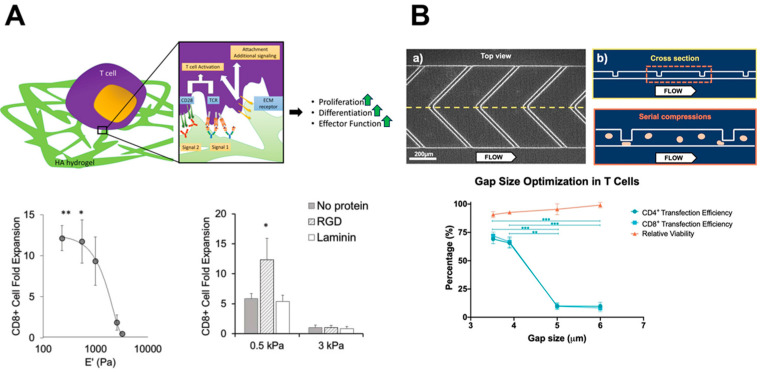
Bioengineering systems improve cancer immunotherapy strategies. Bioengineering systems added into the immunotherapy pipeline can offer enhanced propagation of CD8+ T cells and increased CAR-T generation. Cell culture platforms containing native ECM components that are tuned to physiologically relevant stiffnesses increase CD8+ T cell propagation. This study reveals that changes in substrate stiffness and ECM composition influence T cell propagation (adapted from [141] * *p* < 0.05, ** *p* < 0.005, *** *p* < 0.0005) (**A**). Microfluidic systems can be designed to increase CAR-T transfection efficiency via serial physical compression, called mechanoporation. Top and cross-sectional views are presented in (a) and (b), respectively. This high-throughput strategy elevates transfection efficiency while maintaining cell viability, offering marked improvements that can be easily integrated into the CAR-T workflow (adapted from [132]) (**B**).

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
