# Peer review of "Biophysical Control of the Glioblastoma Immunosuppressive Microenvironment: Opportunities for Immunotherapy"

_bioengineering, 2024, doi:10.3390/bioengineering11010093_

Round 1

Reviewer 1 Report

Comments and Suggestions for Authors

Dear Editor,

I read this article with a great interest. This article reminded me of Galileo Galile's statement: The universe cannot be read until we have learnt the language and become familiar with the characters in which it is written. It is written in mathematical language, and the letters are triangles, circles and other geometrical figures, without which means it is humanly impossible to comprehend a single word. When thought with a similar philosophy, it can be said that understanding glioblastoma or other malignancies is possible by analyzing them with mathematical and physical laws. When the hardware and software changes that carry molecules, cells or tissue communities that want to establish a new order to the process of hypertrophy, hyperplasia or cancer are explained with differential equations, the past, present and future of the tumor can be predicted. Because the laws of biology and chemistry are no longer sufficient to understand the normal and abnormal laws of life. In this article written about glioblastomas, examining glioblastomas with biophysical laws heralds that there will be very successful revolutions in the diagnosis, treatment and prognosis of these tumors in the future. The statement in Figure-1: "The biophysical properties of glioblastoma cells will affect immune cell-cancer cell interactions..." has opened a huge era in terms of a clearer understanding of glioblastomas. It can be said that; The analysis of microscope data with the same laws will make a greater change in the understanding of biology than the change that the analysis of telescope data with the laws of mathematics and physics has made in the understanding of the universe. Writers have opened the door to this great revolution of the future. I congratulate the authors and the team who accepted and published this article.

Kind regards ...

Author Response

We thank the reviewer for their great enthusiasm for this review. We share the same sentiment as the reviewer and greatly appreciate their kind words. Our hope is that this work helps to increase the recognition that perspectives of biophysics and mechanobiology are a critical framework for investigation of the tumor microenvironment.

Reviewer 2 Report

Comments and Suggestions for Authors

In this manuscript, Teer et al review how biophysical properties of the tumor microenvironment (TME) control its immunosuppressive behaviour and how it could be targeted for immunotherapy for gioblastoma. Unfortunately, the review is full of verbosity with little details on molecular mechanisms involved. There is little discussion as to what changes in the TME make it immunosuppressive and how they induce immunosuppression and promote tumor progression and metastasis. It lacks discussion on which mechano-sensors on tumor cells are implicated. Furthermore, the Figures add little to our understanding; their legends being completely incomprehensible.

Author Response

We thank the reviewer for their perspective. We agree that this review does not comprehensively describe the immunosuppressive TME including immunosuppressive cells, stromal cells, and secreted factors; however, the scope of this work focuses exclusively on the biophysical compartment and its role in regulating immunosuppressive. Although a comprehensive discussion is warranted, we believe that there are several excellent reviews in this realm over the past couple of years [1-3], but there are much fewer works focusing on the biophysical compartment of the immunosuppressive TME. We detail the biophysical changes that occur throughout tumor development and highlight several physical inputs, including substrate stiffness, intratumoral pressure, and confinement, that promote immunosuppressive activity through mechanotransductive mechanisms. To address the reviewer's critique on a lack of mechanism, we have added additional reports that detail more mechanistic descriptions of cell-ECM interactions, including mechano-activation of the FAK --> MRTF and PDGF-R axis, that regulate immunosuppression. Additionally, we have updated and modified Figure 2 to more clearly illustrate the significance of leveraging mechanobiological mechanisms via engineered systems to improve the immunotherapy pipeline. We believe that our work provides a unique contribution to the consideration of immunosuppression in the TME.

[1] Labani-Motlagh A, Ashja-Mahdavi M, Loskog A. The tumor microenvironment: a milieu hindering and obstructing antitumor immune responses. Frontiers in immunology. 2020 May 15;11:940.

[2] Tie Y, Tang F, Wei YQ, Wei XW. Immunosuppressive cells in cancer: mechanisms and potential therapeutic targets. Journal of Hematology & Oncology. 2022 May 18;15(1):61.

[3] Barnestein R, Galland L, Kalfeist L, Ghiringhelli F, Ladoire S, Limagne E. Immunosuppressive tumor microenvironment modulation by chemotherapies and targeted therapies to enhance immunotherapy effectiveness. Oncoimmunology. 2022 Dec 31;11(1):2120676.

Reviewer 3 Report

Comments and Suggestions for Authors

The manuscript by Teer et al. provides a comprehensive review of different biophysical methods employed in the study of tumors. The authors directed their attention towards studying the human glioblastoma model due to its distinct characteristics, both in terms of the tumor itself and its surrounding microenvironment (TME).

This review examines two main areas of interest. The first is the nature of brain tumors, while the second focuses on the biological and physical characteristics of TME. To this aim, the authors introduced the concept of mechano-transduction, which forms the basis of their presentation. It is worth noting that this concept has not yet been widely disseminated within the immunological community.

The authors begin their analysis by examining the most well-known components of extracellular matrix (ECM) proteins. Subsequently, they introduce analytical techniques to study components such as ECM mechanics and interstitial fluid flow. These aspects are typically overlooked when studying solid tumors in other tissues, particularly in blood. Yet these features likely influence most of the physical interactions taking place between the ECM and tumors. These interactions become even more complex due to the presence of soluble factors and cell subsets involved in immune defense. The authors also consider the possibility that this environment and such interactions may modify and rewire macrophages and lymphocytes. Figure 1 provides a clear representation of the key steps involved in the cross talk between tumors and their microenvironment.

Figure 2 offers an overview of additional areas related to bioengineering systems, including Biomaterials and Microfabrication. However, it appears targeted towards an expert audience, as it lacks the level of clarity and depth of Figure 1.

The manuscript is well written and contains a comprehensive and up-to-date bibliography.
